# Alternative Splicing of Functional Genes in Plant Growth, Development, and Stress Responses

**DOI:** 10.3390/ijms26125864

**Published:** 2025-06-19

**Authors:** Guan Liu, Hanhui Wang, Huan Gao, Song Yu, Changhua Liu, Yang Wang, Yan Sun, Dongye Zhang

**Affiliations:** 1College of Advanced Agriculture and Ecological Environment, Heilongjiang University, Harbin 150080, China; 2019057@hlju.edu.cn (G.L.); 2022112@hlju.edu.cn (H.G.); liuchanghua@hlju.edu.cn (C.L.); 2004084@hlju.edu.cn (Y.W.); 2004094@hlju.edu.cn (Y.S.); 2State Key Laboratory of Tree Genetics and Breeding, College of Forestry, Northeast Forestry University, Harbin 150040, China; 18846938745@163.com (H.W.); yusong@nefu.edu.cn (S.Y.)

**Keywords:** abiotic stress, alternative splicing (AS), biotic stress, functional genes, plant growth and development

## Abstract

In plants, alternative splicing (AS) is a crucial post-transcriptional regulatory mechanism that generates diverse mature transcripts from precursor mRNA, with the resulting functional proteins regulating a wide range of plant life activities. The regulation of AS is intricate and complex, playing pivotal roles in controlling plant biological processes like seed germination, flowering time control, growth, and development, as well as responses to abiotic and biotic stresses. The regulation of AS is a multilayered and intricately coordinated network system, primarily involving two core components: cis-regulatory elements and trans-acting factors on pre-mRNA. The precise execution of AS relies on the splicing factors by recognizing cis-elements to modulate splice site selection. Regulated by their own sequence variation, environmental cues, and identification of different spliceosomes, functional genes enable AS to achieve precise spatiotemporal regulation, thereby allowing plants to dynamically respond to developmental signals and environmental challenges. Here, we provide a comprehensive overview of AS patterns, functional genes, and splicing factors undergoing AS and its regulatory mechanisms during different processes, highlighting how AS-mediated gene regulation contributes to plant development and stress response, and offering potential strategies for improving plant adaptation by manipulation of AS-regulated genes.

## 1. Introduction

In plants, alternative splicing (AS) is widespread and plays a vital role in regulating growth and development, flowering time, circadian clock, and responses to abiotic stresses such as drought, high temperature, and salinity. Early studies first reported AS events in spinach (*Spinacia oleracea*) and Arabidopsis (*Arabidopsis thaliana*) [1], and subsequent research has extensively validated its occurrence in various plant species, including monocotyledons like rice (*Oryza sativa*), maize (*Zea mays*) and *Brachypodium distachyon*; as well as dicotyledons such as *Brassica oleracea*, kiwifruit (*Actinidia chinensis*) [2,3,4,5,6]. Recent high-throughput transcriptome sequencing has revealed that over 60% of multi-exon genes in plants undergo AS, exhibiting tissue specificity, developmental stage specificity, or environmental inducibility [5,6,7].

AS events in plants primarily include five types: exon skipping, alternative 5′ splicing, alternative 3′ splicing, intron retention, and mutually exclusive exons (Figure 1). It precisely recognizes and excises introns from pre-mRNA, then ligates the exons to generate mature mRNA, thereby ensuring the accuracy of gene expression (Figure 2). The regulation of AS is a multilayered and intricately coordinated network system [8], primarily involving two core components: *cis*-regulatory elements and *trans*-acting factors on pre-mRNA. Notably, sequence variations within genes—such as point mutations or indels—can alter these *cis*-elements, thereby disrupting existing splice sites or creating novel ones, ultimately leading to changes in splicing patterns. *Trans*-acting factors, on the other hand, are typically RNA-binding proteins that interact with these *cis*-elements to promote or inhibit splice site usage, including SR proteins and the hnRNP family (Figure 2). Additionally, this regulatory network also integrates factors such as histone modifications, DNA methylation, chromatin environment, and RNA structures [9], which are not mentioned in this study.

AS has shaped the evolution of complex regulatory networks governing gene expression and variation, enabling the regulation of key genes involved in plant stress responses [10]. Many studies have separately reported that AS of diverse functional genes involved in the transcriptional regulation of higher plants could regulate plant development, and response to both biotic and abiotic stresses, providing an effective regulatory mechanism for plants to rapidly adapt to environmental changes [11,12,13,14,15,16,17]. However, existing reviews mostly focus on a single developmental stage or specific stresses, lacking a systematic integration of the overall role of AS in plant growth, development, and responses to multiple stresses. To address this gap, this study emphasizes recent research progress in these areas.

Here, we focus on how cis-regulatory elements and trans-acting factors mediate alternative splicing regulation of functional genes and spliceosome components during plant growth, development, and stress responses, providing a systematic integration of AS mechanisms. We conducted a systematic literature search using databases such as PubMed, Web of Science, and Google Scholar, with a primary focus on studies published in the past five years and supplemented with earlier key references as background. Initial screening was based on titles and abstracts focused on the theme of AS regulation during plant growth, development, and responses to environmental stresses, followed by a full-text review to determine inclusion in the review. This study aims to deepen our understanding of the roles of AS in key physiological processes by highlighting AS regulation of functional genes and splicing factors during plant growth, development, and responses to biotic and abiotic stresses, while also offering perspectives for future research.

## 2. Key Genes Undergoing Alternative Splicing During Plant Growth and Development

### 2.1. Seed Development

The timing of seed germination determines the successful establishment of plants, with seed dormancy serving as a key factor controlling germination potential. In Arabidopsis, the plant hormone ABA (abscisic acid) regulates seed maturation and dormancy by inhibiting germination and reserve mobilization [18,19], and previous studies have demonstrated that AS plays a crucial role in this process by modulating key ABA signaling components. For example, the DOG1 (*DELAY OF GERMINATION 1*) gene regulates seed dormancy depth by producing multiple transcript variants; the spliceosome core component SmEb modulates AS of the HAB1 (*HYPERSENSITIVE TO ABA1*) gene to balance ABA responses; and splicing factor SUA (SUPPRESSOR OF ABI3-5) influences seed maturation by regulating AS of *ABI3* (*ABSCISIC ACID IN SENSITIVE3*), thereby affecting seed maturation [20,21,22].

More recently, DRT111 (DNA-DAMAGE REPAIR/TOLERATION PROTEIN111) has been identified as a novel and critical splicing factor by modulating the splicing of ABI3 pre-mRNA during seed development (Figure 3A). DRT111 suppresses the accumulation of the non-functional ABI3-β isoform and ensures proper expression of the functional ABI3-α, and loss of DRT111 leads to increased ABI3-β levels, causing ABA hypersensitivity and impaired germination. In coordination with SUA, DRT111 precisely regulates ABI3 splicing, ensuring appropriate ABA sensitivity during seed development and germination. Transcriptomic evidence reveals that DRT111 deficiency causes widespread alternative splicing changes in ABA signaling, light response, and RNA processing genes, highlighting its central role as a regulatory hub linking environmental cues with seed fate decisions [23].

In wheat (*Triticum aestivum*), the TaPP2C-a5 gene undergoes AS to produce two isoforms, TaPP2C-a5.1 and TaPP2C-a5.2, which coordinately link the DOG1 and ABA signaling pathways to regulate seed dormancy and germination. Notably, *TaPP2C-a5.2* exhibits seed-specific expression and represents a potential breeding target for regulating seed dormancy, offering a promising approach to improve pre-harvest sprouting resistance [24]. Another study in barley (*Hordeum vulgare*) revealed that CBC (cap-binding complex) participates in the response to ABA by regulating AS. In the *hvcbp20.ab/hvcbp80.b* double mutant, several splicing factors (such as SUA, CDC5, and U2AF35A) are upregulated, leading to altered AS regulation, significant changes in brassinosteroid signaling and enhanced ABA tolerance which facilitated seed germination, suggesting that CBC acts as a key molecular hub linking ABA and BR signaling pathways [25].

### 2.2. Vegetative Growth and Morphogenesis

When a seed germinates, the hypocotyl forms a curved structure at its apex called the apical hook, which protects the crucial shoot apical meristem from physical damage in the soil. The PIN (PIN-FORMED) family encodes polar auxin transporters, which play a role in maintaining the activity of the root apical meristem from the early seedling stage [26]. Previous studies have reported that alternative 5′SS at the end of the first exon of *At-PIN7* generated two evolutionarily conserved transcripts, PIN7a and PIN7b, which act antagonistically (Figure 3B). These two isoforms differ mainly by the presence or absence of a conserved four–amino acid microdomain, GETK. This subtle sequence variation, without altering polarity localization or transport activity, significantly affects their dynamic behavior on the plasma membrane and intracellular trafficking mechanisms. PIN7a exhibits higher membrane stability and slower recovery rates, facilitating stable polarity and efficient auxin transport; whereas PIN7b shows faster mobility and greater regulatory flexibility. Moreover, these two isoforms can form dimers or higher-order complexes, mutually modulating their membrane dynamics and collectively influencing the establishment and regulation of auxin gradients. This AS event provides a molecular basis for PIN7’s flexible response to different developmental stages or environmental conditions, and contributes importantly to phenotypes such as embryogenesis, proper apical hook formation, and tropic growth in seedlings [27].

### 2.3. Flowering and Development

Flowering represents a crucial developmental switch from vegetative growth to the reproductive phase in plants, governed by an intricate regulatory network. The core flowering pathways involving FT (FLOWERING LOCUS T) and FLC (FLOWERING LOCUS C) have been well characterized [28]. Notably, multiple flowering-related genes in Arabidopsis have been found to undergo AS, including *FLM*/*MAF1* (*FLOWERING LOCUS M/MADS ASSOCI ATED FLOWERING1*), *CO* (*CONSTANS*) and *FCA* (*FLOWERING CONTROL LOCUS A*) and *FLC* [29,30,31,32,33]. Studies of these classic genes have laid the foundation for understanding how AS generates protein isoforms with distinct functions to finely regulate the molecular mechanisms controlling flowering time in plants. For instance, the *FLM* gene generates temperature-dependent isoforms *FLM-β* and *FLM-δ*, which competitively interact with the floral repressor SVP; whereas the *CO* gene produces full-length *CO-α* and truncated *CO-β*, with *CO-β* inhibiting *CO-α* activity by forming nonfunctional heterodimers [31,33].

Recently, AS-mediated flowering regulation of other functional genes has recently been reported in diverse species. In pear (*Pyrus* sp.), the circadian clock regulator *EARLY FLOWERING 3* (*ELF3*) has been identified as a key factor involved in the fine-tuning of flowering time through AS mechanisms (Figure 3C). *ELF3* generates two transcript isoforms with opposing functions via AS: *ELF3α*, which represses flowering, and *ELF3β*, which alleviates this repression. *ELF3β* is produced from an intronic alternative promoter and retains part of the intron sequence after splicing (Figure 3). Deletion of the key intronic region significantly reduces ELF3β expression, thereby enhancing ELF3α activity and resulting in delayed flowering. The functional conservation of ELF3β has also been demonstrated in multiple species, including apples, soybeans, and tomatoes, suggesting a broadly conserved flowering regulatory mechanism across plants [34]. In citrus (*Citrus* spp.), the *CiFD* gene generates two distinct protein variants through AS—*CiFDα* and *CiFDβ*—which are differentially regulated by low temperature and drought stress respectively to modulate flowering timing [15]. Furthermore, cross-species analyses reveal conserved AS patterns in flowering regulators: *PaFT* (21 variants) in *Platanus acerifolia*, *PtFCA/PtFLC* isoforms in trifoliate orange (*Poncirus trifoliata*), *PeCOL13α/β* in *Phyllostachys edulis*, and *BjuAGL18-1* transcripts in *Brassica juncea* [14,35,36,37,38], highlighting the prevalence and diversity of AS-mediated fine-tuning mechanisms in plant flowering time regulation.

Some studies have also demonstrated the critical roles of AS in organ development. A *G-to-A* single nucleotide mutation in the cotton (*Gossypium arboreum*) *TFL1* (*TERMINAL FLOWER 1*) gene alters the 5′ SS at the boundary of the first exon and intron, causing aberrant splice site selection and a truncated protein missing 31 amino acids, leading to a complete loss of function (Figure 3D) [39]. This mutation affects the fate determination of the shoot apical meristem and plant growth, and the identified new RNA splicing site could provide new insights into the post-transcriptional regulatory mechanisms underlying plant architecture and development.

### 2.4. Maturation and Senescence

Fruit ripening encompasses multiple coordinated processes including phytohormone signaling, cell wall degradation, biosynthesis of flavor/aroma compounds, synthesis/degradation of pigments, and metabolic processes, all of which could be modulated by AS of relevant genes [40]. In banana (*Musa acuminata*), AS of *MaMYB16L* generates a self-regulatory circuit controlling fruit ripening, where downregulation of the full-length isoform (*MaMYB16L*) and upregulation of the truncated variant (*MaMYB16S*) fine-tune starch degradation-related gene expression to promote maturation (Figure 3E) [41]. In blood orange (*Citrus sinensis* cv ‘Tarocco’), AS of *CsTT8* (*CsTRANSPARENT TESTA 8*) generates three isoforms, where the relative *TT8/Δ15-TT8* transcript ratio determines anthocyanin production in both pulp and peel tissues. Notably, the *Δ15-TT8* isoform negatively regulates *TT8* expression through a feedback mechanism, thereby precisely modulating fruit pigmentation accumulation [42]. Furthermore, in tomato (*Solanum lycopersicum*), the regulatory mechanism of *SlIDI1* (*Isopentenyl diphosphate isomerase 1*) depends on AS during fruit ripening, where alternative transcription initiation and splicing generate both long and short transcripts; only the long transcript participates in carotenoid biosynthesis within petals and anthers [43]. Contrasting with the red-fruited cultivars, the *yellow flesh tomato2* (*S. lycopersicum* var. *cerasiforme*) exhibits a novel *SlPSY1* (*PHYTOENE SYNTHASE 1*) splicing pattern that produces an elongated transcript variant (*LT-YFT2*), which alters carotenoid accumulation and ultimately leads to yellow fruit pigmentation [44]. Genome-wide transcriptomic analyses have further revealed extensive differentially alternative splicing (DAS) events and differentially spliced genes (DSGs) during fruit ripening in plants such as grape (*Vitis vinifera*), coffee (*Coffea arabica*) and *Capsicum* species [45,46,47].

During plant senescence, diverse internal and external signals, including nutrient/hormonal signals, osmotic stress, temperature changes, and light conditions, interact with age to coordinately regulate leaf senescence [48]. Some studies have highlighted the pivotal roles of AS in this process across model species, including Arabidopsis, rice (*Oryza sativa*), and poplar (*Populus tomentosa*) [49,50]. In rice, both transcript variants of *ONAC054* (*ONAC054α* and *ONAC054β*) play essential roles in ABA-induced leaf senescence [49]. In poplasr autumn senescence, the *PtRD26* regulator undergoes natural intron retention to produce *PtRD26^IR^*, an isoform specifically enriched in senescing leaves that negatively regulates *PtRD26*-driven senescence [51].

As demonstrated above, these cases demonstrate how distinct isoforms generated through AS exhibit tissue- and developmental stage-specific activities and biological functions, playing pivotal roles throughout plant growth and developmental processes. In Figure 3, one gene was randomly selected for each process as an illustrative example to visualize the functional relevance of AS, while Table 1 provides a more comprehensive summary of genes and AS events discussed in this study.

## 3. Alternative Splicing of Stress-Regulatory Genes During Plant Adaptation

### 3.1. Temperature Stresses

AS displays strong temperature sensitivity, where both cold and heat stresses affect transcriptome stability and induce distinct isoforms. HSFs (Heat shock transcription factors) are pivotal regulators of elevated temperature or heat shock responses, with severe heat-induced AS being a conserved regulatory feature of *HSF* genes across diverse species including Arabidopsis, rice, alfalfa (*Medicago sativa*), wheat (*Triticum aestivum*), lily (*Lilium* spp.), and genus *Potamogeton* [16,59,60,61,62,63,64,65].

In wheat, TaHSFA6e undergoes temperature-dependent AS under heat stress conditions, generating two major transcripts: *TaHSFA6e-II* and *TaHSFA6e-III* (Figure 4A). TaHSFA6e-III contains an additional 14 amino acid peptides introduced by AS at the C-terminus of the protein, which significantly enhances the transcriptional activation ability of TaHSFA6e-III on three downstream TaHSP70 genes. Through this splicing regulation, the TaHSFA6e–TaHSP70s module contributes to improved thermotolerance in wheat [16]. Additionally, another heat-responsive gene, *RDM16* (*RNA-DIRECTED DNA METHYLATION16*), produces two alternatively spliced transcripts—*RDL* (*RDM16-LONG*) and *RDS* (*RDM16-SHORT*), where RDS could enhance RDL-mediated heat resistance [52].

For cold stress, the *ICE* (*INDUCER OF CBF EXPRESSION*) -*CBF* (*C-REPEAT-BINDING FACTOR*) -*COR* (*COLD-REGULATED*) pathway serves as one of the key mechanisms mediating plant cold tolerance by regulating the expression of low-temperature-related genes [13]. Existing studies have reported that CBF not only regulates the expression of *COR* (*COLD-REGULATED/COLD-RESPONSIVE*) but also interacts with multiple spliceosome-associated components to modulate AS efficiency, particularly SKIP (SKI-INTERACTING PROTEIN) (Figure 4B). This interaction promotes the formation of liquid-liquid phase-separated nuclear condensates of SKIP, which may enhance the association of the SKIP–spliceosome complex with specific target RNAs, thereby regulating AS events of particular genes. In addition, CBFs may further influence cold-induced AS by interacting with other spliceosome-associated proteins or by modulating the expression of splicing regulators, thereby facilitating rapid transcriptome reprogramming in response to cold stress [13].

Currently, PacBio isoform sequencing (Iso-Seq) provides a direct and effective approach for obtaining full-length transcript sequences, allowing for precise annotation of alternative splicing events. In trifoliate orange (*Citrus trifoliata*), a large number of AS events occur under cold stress, with IR being the predominant type. Thousands of differentially spliced genes (DSGs) were identified, which are widely involved in biological pathways such as basic metabolism, stress response, photosynthesis, and carbon metabolism. Notably, several key transcription factors, including ERF, MYB, and bZIP families, not only showed upregulated expression under cold stress but also underwent AS. This suggests that AS contributes to cold adaptation by modulating the expression and function of these transcription factors, highlighting its central role in the cold signaling network [66].

### 3.2. Drought Stress

As a major abiotic stress, drought severely impacts crop productivity, prompting extensive investigations into its responsive mechanisms in various plant systems, ranging from model organisms to economically important crops. In rice, *OsbHLH59* produces two functionally complementary transcripts, *OsbHLH59.1* and *OsbHLH59.2*, through ABA-dependent AS, which respectively regulate plant growth and drought stress response (Figure 4C). Under normal conditions, *OsbHLH59* primarily exists as OsbHLH59.1, promoting the expression of development-related genes and facilitating growth; under drought stress, ABA signaling induces AS to generate OsbHLH59.2, which forms heterodimers with OsbHLH59.1 to suppress growth-related gene expression while activating drought-responsive target genes, thereby achieving a dynamic balance between growth and drought tolerance [12]. In maize, the expression levels of *ZmCCA1* splice variants are influenced by photoperiod, tissue type, and drought stress, with *ZmCCA1.1* being more effective in enhancing drought tolerance in transgenic Arabidopsis plants [53]. In wheat, *TaDREB3* produces three isoforms (*TaDREB3-I*, *-II*, and *-III*), among which both *TaDREB3-I* and *TaDREB3-II* are induced by various abiotic stresses including drought, salt, and high temperature, but only *TaDREB3-I* could significantly enhance the survival rate of transgenic Arabidopsis under these stress conditions [54].

Additionally, genome-wide drought-responsive AS patterns have also been reported. For instance, drought-induced AS patterns vary significantly among drought-tolerant varieties in linseed (*Linum usitatissimum*), and drought-responsive AS in soybean (*Glycine max*) directly or indirectly modulates root responses [67,68].

### 3.3. Salt Stress

Comparative transcriptomics reveals salt-induced alternative splicing diversity: wheat exhibits 11,141 significant AS alterations, and foxtail millet (*Setaria italica*) displays 2078 specific AS events under different salt treatment periods [17,69]. In Arabidopsis, *SRAS1* (*Salt-Responsive Alternatively Spliced gene 1*) gene generates two functionally distinct splice variants under salt stress conditions: SRAS1.1 and SRAS1.2 (Figure 4D). *SRAS1.1* is the full-length transcript containing a complete RING domain and possesses E3 ubiquitin ligase activity, enabling it to recruit ubiquitin molecules to tag target proteins. In contrast, SRAS1.2 retains the first intron, resulting in a premature termination codon that produces a truncated protein lacking the RING domain and thus losing ubiquitin ligase activity. Under normal growth conditions, SRAS1 mainly produces the SRAS1.2 isoform, which competes for binding with the CSN5A (COP9 signalosome 5A) protein, protecting CSN5A from degradation and promoting its stability, thereby supporting plant growth and development. Under salt stress conditions, AS regulation leads to a significant upregulation of SRAS1.1 expression. The SRAS1.1 protein enhances the ubiquitination of CSN5A, promoting its degradation via the 26S proteasome pathway, and regulating intracellular ROS concentration, thereby activating salt stress tolerance mechanisms [55].

Another study reported that AS plays a critical role in the functional diversity and stress adaptability of the *CdDHN4* (*DEHYDRIN*) gene in bermudagrass (*Cynodon* spp.) [56]. This gene generates two splice variants through AS, *CdDHN4-L,* and *CdDHN4-S*, which encode YSK2-type dehydrin proteins either containing or lacking the Φ-segment, respectively. Functional analyses revealed that both splice variants significantly enhanced transgenic Arabidopsis tolerance to salt, drought, osmotic, and low-temperature stresses, but differed in their specific response mechanisms: *CdDHN4-S* showed superior capacity for reactive oxygen species (ROS) scavenging and osmotic stress response, whereas *CdDHN4-L* was more effective under cold and drought conditions. These findings demonstrate that AS diversifies the functional roles of a single gene, allowing plants to finely tune their stress responses and enhance adaptability to complex and varying environmental challenges (Figure 4).

In addition, although recent reports on heavy metal stress and other abiotic stresses are relatively fewer compared to those mentioned above, AS has also been recognized for its broad role in regulating plant environmental adaptability. Proteogenomic analysis revealed that AS plays a key role in the response of triploid poplar (*Populus trichocarpa*) to lead (Pb) stress. Under Pb stress, AS patterns are significantly altered, characterized by an increased use of nonconventional splice sites and the upregulation of splicing factors associated with Pb response. The PtHSP70 gene has been found to undergo AS to produce two spliced isoforms—PtHSP70-AS1 and PtHSP70-AS2—regulated by the core splicing factor PtU1-70K (Figure 4E). Overexpression of either isoform in poplar and Arabidopsis significantly enhances Pb tolerance, while PtHSP70-AS2 exhibits approximately 10 times higher binding Pb(II) capacity than PtHSP70-AS1, showing higher expression under Pb stress, effectively promoting Pb(II) extrusion, reducing cellular toxicity, and thereby enhancing plant Pb tolerance [7].

### 3.4. Biotic Stress

With the advancement of omics technologies, research on AS in plants responding to biotic stresses, especially pathogen infection, has gradually increased. Pathogen infection induces extensive AS of numerous plant genes, and various effector proteins from pathogens can directly target host spliceosome components, thereby controlling the splicing of plant immunity-related genes. For example, the pathogen *Exserohilum turcicum* secretes the effector protein EtEC81 (*Exserohilum turcicum* effector 81), which interacts with the maize spliceosome-associated protein ZmEIP1 (MAIZE EtEC81-INTERACTING PROTEIN 1), thereby reprogramming the maize’s AS process to activate plant immune responses (Figure 4F). ZmEIP1 could bind to multiple spliceosome components, participating in AS regulation and positively modulating plant immunity; transiently overexpression of ZmEIP1 or EtEC81 in maize alters 119 shared AS events, affecting cellular functions and ultimately enhancing defense capacity against the pathogen [11].

Additionally, in the tomato cultivar Heinz 1706, the effector protein RipP2 from *Ralstonia solanacearum* acetylates a conserved lysine residue (K132) in the tomato splicing factor SR34a, thereby regulating SR34a-mediated AS of immune-related genes, which affects 1386 differential alternatively spliced events across 1023 genes (Figure 4F). This regulation particularly promotes IR events in defense genes such as RBP, SNR, ER68, and U2AF65C, leading to the production of truncated, non-functional proteins, which suppresses plant immune responses and reduces disease resistance [57]. In apple (*Malus domestica*), MdMYB6-like has been found to have three AS isoforms. Among them, MdMYB6-like-β shows a positive response to infection by the *Alternaria alternata* apple pathotype, and its overexpression increases leaf lignin content and enhances pathogen resistance in apple fruit callus tissues [58]. Although current research lacks extensive direct evidence, it is reasonable to infer that plants also widely employ AS to dynamically adjust their defense strategies in response to insect herbivory and nematode attacks, given the broad involvement of AS in plant responses to pathogen infection.

Figure 4 illustrates representative examples of genes undergoing AS in response to various abiotic and biotic stresses. For each type of stress, one gene was randomly selected to demonstrate the presence of stress-responsive AS events, highlighting the widespread and diverse roles of AS in environmental adaptation.

## 4. Conclusions

AS plays an indispensable role in plant development by dynamically modulating transcriptome profiles in response to environmental cues. Through generating distinct transcript isoforms, AS enables plants to fine-tune gene expression and adapt to changing external conditions [18,19,70,71]. The emergence of AS in plants is driven both by external environmental stimuli and internal regulatory factors. Functional genes generate different AS isoforms due to sequence variation, and splicing factors recognize specific cis-elements and guide splice site selection, which can be modulated by developmental cues and stress signals (Figure 2). In this study, we summarize the regulatory roles of functional gene AS events in plant development (including seed dormancy/germination, vegetative growth, flowering, fruit ripening, and senescence), abiotic stress responses (heat, cold, drought, salt, and heavy metal stresses) and biotic stress. We highlight key genes with splicing variations identified across various plant species and compile a wide range of representative AS isoforms along with their potential functions (Table 1, Figure 3 and Figure 4). While focusing primarily on functionally spliced genes in these processes (with limited discussion of splicing factors), we note that comprehensive analyses of how splicing factors orchestrate these regulations have been detailed in other reviews [10,72,73].

Plants possess distinct AS mechanisms that are more complex than those in animals, owing to their unique genomic features (including intron size, nucleotide composition, and branch point sequences) and evolutionary genome duplication events [74]. By gaining a deeper understanding of the AS patterns in plant development and stress responses, new biotechnological strategies can be developed to improve crops. Emerging technologies are now shedding light on the widespread occurrence of stress-induced AS in plants. CRISPR-based editors, in particular, offer a powerful and efficient approach to investigating the functions of specific splice variants, even when dealing with highly complex AS events. Precise editing of functional genes and splicing factors can alter the AS patterns of specific target genes, thereby affecting protein diversity and function. This, in turn, can be used to improve important agronomic traits such as crop stress resistance, yield, and quality. However, achieving precise splicing remains a significant challenge in the context of multiple coexisting splice sites and overlapping or antagonistic isoform functions. Future research should focus on further developing high-precision splicing regulation tools, multi-omics approaches combining epigenetics and transcriptomics, and integrating computational modeling with biology approaches to enable precise control of splicing.

While current research often focuses on analyzing putative protein products to understand AS transcript functions, a more expansive perspective that considers potential RNA-centric regulatory roles needs to be adopted. Novel strategies will be critical for comprehensively analyzing, evaluating, and characterizing the functional implications of splice variants of interest [71]. Although this study successfully identified a large number of candidate functional genes exhibiting AS during plant development and stress responses (Table 1) and revealed the widespread nature of AS events, this is only the first step toward understanding the complex regulatory mechanisms of AS. Current understanding of splicing regulatory initiation signals, the specific interactions between *cis*-acting splicing elements and *trans*-acting factors, as well as the spatial interplay among chromatin accessibility and RNA structure, remains insufficient, which seriously limits the promotion and innovative breakthroughs of the AS mechanism. Future research should focus on mechanistically dissecting the core components of AS events and their signal transduction pathways. For example, investigating how natural environmental signals broadly regulate the selective binding of the spliceosome to target genes, how multiple AS events are regulated through the activation of specific RBPs, SRs, or other auxiliary factors, and how to precisely control splice sites to generate novel AS events. Additionally, it is also essential to deeply analyze the causal relationships between epigenetic regulation and AS, as well as their synergistic roles in tissue-specific splicing events. Moreover, splicing factors themselves often undergo AS in plants, forming multi-level positive or negative feedback regulatory loops. Exploring the dynamic behavior of such loops will be a critical direction for future AS network modeling and the prediction of regulatory targets.

## Figures and Tables

**Figure 1 ijms-26-05864-f001:**
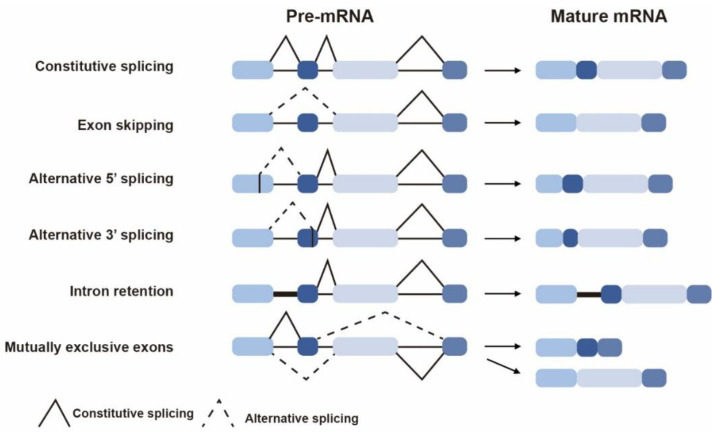
The major types of alternative splicing events. Different colored boxes represent different exons; the black straight lines between the boxes represent introns; the black zigzag lines indicate constitutive splicing sites, and the dashed zigzag lines indicate alternative splicing sites.

**Figure 2 ijms-26-05864-f002:**
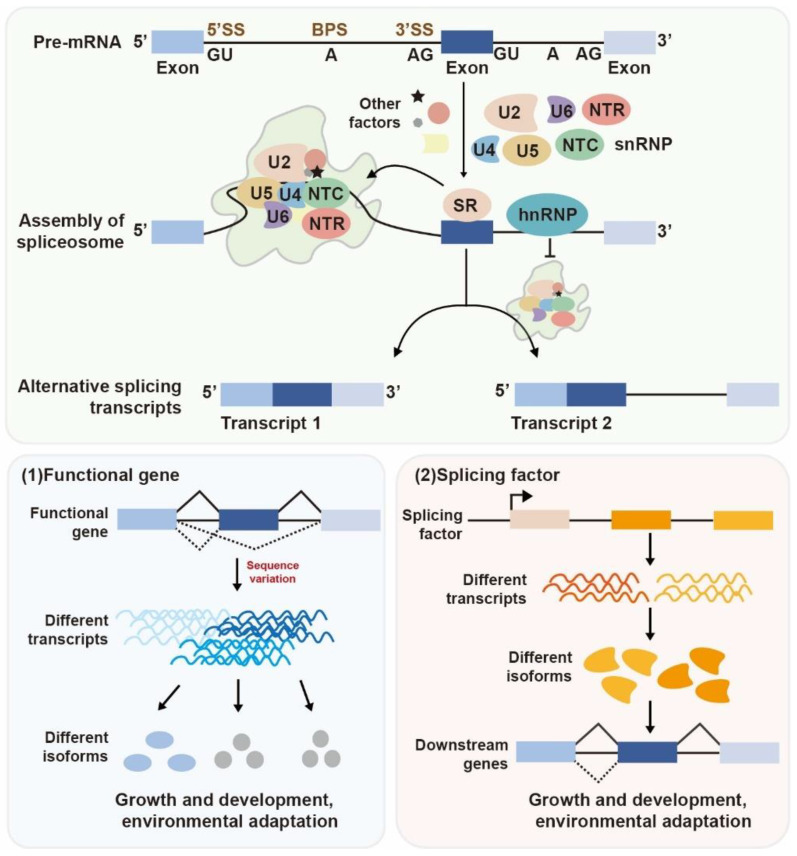
A schematic diagram mechanism of AS regulation. AS can be influenced by two major factors: *cis*-element variations within functional genes and *trans*-acting splicing factors: (1) Sequence variations in functional genes could change splice sites, leading to the generation of multiple transcript isoforms with distinct functions. (2) Many splicing factors undergo AS themselves, which in turn modulates the splicing of downstream target genes. Exons are represented by blue rectangles, and introns are represented by black lines. Different shapes and colors represent different factors. BPS, branch point sequence; hnRNP, heterogeneous nuclear ribonucleoprotein; NTC, NineTeen Complex; NTR, NTC-related complex; pre-mRNA, precursor messenger RNA; snRNA, small nuclear RNA; snRNP, small nuclear ribonuclear protein; SR, serine/arginine-rich RNA-binding protein; SS, splice site.

**Figure 3 ijms-26-05864-f003:**
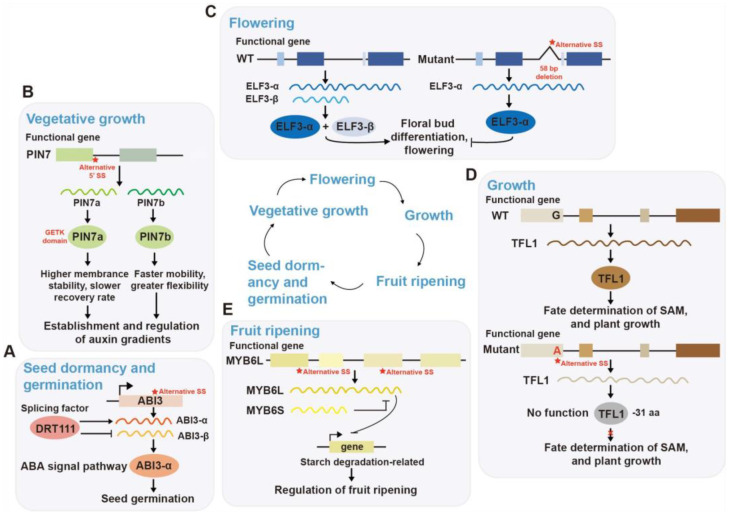
AS of functional genes during plant growth and development mentioned in this study. Boxes represent exons, wavy lines represent transcripts, and circles represent proteins. (**A**) Seed dormancy and germination: The splicing factor DRT111 promotes the generation of the ABI3-α isoform through AS, thereby activating ABA signaling and regulating seed germination. (**B**) Vegetative growth: The *PIN7* gene undergoes alternative 5′ splice site selection to generate two isoforms, *PIN7a* and *PIN7b*. The presence or absence of the GETK motif leads to distinct membrane dynamics, with PIN7a showing higher stability and PIN7b exhibiting greater mobility. These isoforms coordinate the establishment and regulation of auxin gradients. (**C**) Flowering: The *ELF3* gene produces two isoforms, ELF3-α and ELF3-β, through AS. In wild-type plants, both isoforms promote floral bud differentiation and flowering, while in mutants with a 58 bp deletion, only ELF3-α is produced, impairing the regulatory balance. (**D**) Growth: The *TFL1* gene regulates shoot apical meristem (SAM) fate. A single nucleotide mutation at the splicing site results in a truncated, non-functional TFL1 protein lacking 31 amino acids, thereby affecting SAM determination and plant growth. (**E**) Fruit ripening: The MYB6L gene produces two isoforms, MYB6L and MYB6S, via AS. These isoforms differentially regulate downstream genes involved in starch degradation, modulating fruit ripening. SS, Splicing site. Each developmental or physiological process is illustrated with one representative example gene to demonstrate the role of AS.

**Figure 4 ijms-26-05864-f004:**
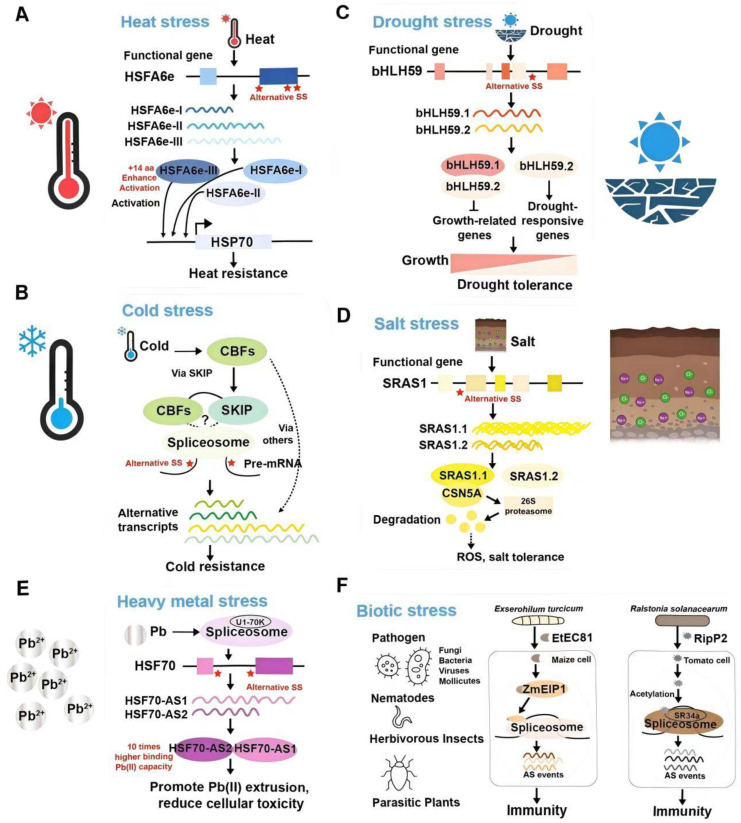
AS of functional genes during abiotic and biotic stresses. Each developmental or physiological process is illustrated with one representative example gene to demonstrate the role of alternative splicing. (**A**) Heat stress: The HSFA6a gene produces multiple isoforms (HSFA6a-I, II, III) through AS. HSFA6a-III can enhance the activation of downstream heat-responsive genes like HSP70, thereby improving thermotolerance. (**B**) Cold stress: CBFs promote AS reprogramming during cold stress by interacting with spliceosome components like SKIP, enhancing the formation of phase-separated nuclear condensates. This targeted modulation of AS facilitates rapid adaptation and improved cold tolerance in plants. (**C**) Drought stress: The bHLH59 gene generates isoforms bHLH59.1 and bHLH59.2 via AS. These isoforms differentially regulate growth-related and drought-responsive genes, coordinating plant growth and drought tolerance. (**D**) Salt stress: SRAS1 undergoes AS to produce SRAS1.1 and SRAS1.2. SRAS1.2, a truncated isoform, stabilizes CSN5A during normal growth, while SRAS1.1 promotes CSN5A degradation under salt stress, thereby enhancing ROS detoxification and salt tolerance. (**E**) Heavy metal stress: Lead exposure influences splicing of the HSF70 gene, producing isoforms HSF70-AS1 and HSF70-AS2. HSF70-AS2, with enhanced Pb(II)-binding capacity, promotes lead extrusion and reduces toxicity. (**F**) Biotic stress: Pathogen effectors (e.g., EtEC81 from *Exserohilum turcicum* and RipP2 from *Ralstonia solanacearum*) target spliceosome components in maize and tomato, respectively, leading to reprogramming of host AS events and activation of immune responses through factors such as ZmEIP1 and SR34. Each panel demonstrates how AS fine-tunes gene expression and protein function, enabling plants to dynamically adapt to specific stress conditions.

**Table 1 ijms-26-05864-t001:** List of genes producing alternatively spliced isoforms during different developmental stages or stress conditions in plants.

Gene Name	Species	Stage/Stress	Type	Isoform	Function	Reference
DOG1	*Arabidopsis thaliana*	Seed development	Alternative splicing site	5 isoforms	Regulates depth of seed dormancy	[20]
HAB1	*Arabidopsis thaliana*	Seed development	Intron retention	HAB1α/HAB1β	Balances ABA responses during seed germination	[21]
ABI3	*Arabidopsis thaliana*	Seed development	Exon skipping	ABI3-α/ABI3-β	Functional vs. non-functional; affects seed maturation	[22,23]
TaPP2C-a5	*Triticum aestivum*	Seed development	Exon skipping	a5.1/a5.2	a5.2 is seed-specific; regulates ABA-DOG1 coordination	[24]
PIN7	*Arabidopsis thaliana*	Vegetative growth	Alternative 5′ splice site	PIN7a/PIN7b	Different membrane dynamics; fine-tune auxin transport	[27]
FLM	*Arabidopsis thaliana*	Flowering	Exon inclusion/skipping	FLM-β/FLM-δ	Competes for SVP binding, temperature-dependent isoforms modulate flowering via SVP interaction	[30,31]
CO	*Arabidopsis thaliana*	Flowering	Truncated transcript	CO-α/CO-β	CO-β forms non-functional dimers to inhibit CO-α activity	[33]
ELF3	*Pyrus sp.*	Flowering	Intron retention & alt. promoter	ELF3α/ELF3β	ELF3β alleviates repression of flowering by ELF3α	[34]
FD	*Citrus spp.*	Flowering	Exon skipping	CiFDα/CiFDβ	Responds to cold/drought to regulate flowering time	[15]
FT	*Platanus acerifolia*	Flowering	Alternative splicing site	21 isoforms	Exhibits conserved AS pattern in flowering regulation	[35]
FCA/FLC	*Poncirus trifoliata*	Flowering	Alternative splicing site	*PtFCA1-3*, *PtFLC1-5*	Modulate flowering time	[36,37]
COL13	*Phyllostachys edulis*	Flowering	Intron retention	PeCOL13α/PeCOL13β	Regulates photoperiod-responsive flowering	[38]
AGL18-1	*Brassica juncea*	Flowering	Truncated transcript	BjuAGL18-1L/BjuAGL18-1S	Involved in flowering regulation	
TFL1	*Gossypium arboreum*	Vegetative/Floral development	Splice site mutation	Full-length/Truncated	Loss-of-function mutant affects meristem fate	[39]
MYB16	*Musa acuminata*	Fruit maturation	Truncated transcript	MaMYB16L/MaMYB16S	Modulates starch degradation via isoform ratio	[41]
TT8	*Citrus sinensis*	Fruit pigmentation	Exon skipping	TT8/Δ15-TT8	Fine-tunes anthocyanin biosynthesis via feedback regulation	[42]
IDI1	*Solanum lycopersicum*	Fruit maturation	Alternative transcription initiation and alternative splicing	Long/short isoforms	Only the long isoform is functional in carotenoid synthesis	[43]
PSY1	*S. lycopersicum var. cerasiforme*	Fruit pigmentation	Alternative *trans*-splicing	*LT-YFT2/YFT2*	Alters carotenoid content in yellow-fruited variants	[44]
ERF4	*Arabidopsis thaliana*	Leaf senescence	Alternative splicing site	ERF4-R/ERF4-A	ERF4-R promotes, ERF4-A suppresses senescence	[50]
NAC054	*Oryza sativa*	Leaf senescence	Exon skipping	ONAC054α/ONAC054β	Both isoforms regulate ABA-induced senescence	[49]
RD26	*Populus tomentosa*	Leaf senescence	Intron retention	PtRD26/PtRD26IR	Negatively regulates senescence in aging leaves	[51]
HSFA6e	*Triticum aestivum*	Heat stress	Exon extension	TaHSFA6e-II/TaHSFA6e-III	AS generates TaHSFA6e-III with enhanced transactivation ability on TaHSP70s, improving thermotolerance	[16]
RDM16	*Arabidopsis thaliana*	Heat stress	Exon skipping	RDL, RDS	RDS enhances RDL-mediated heat resistance via functional interaction	[52]
CBFs	*Arabidopsis thaliana*	Cold stress	Spliceosome	Spliceosome modulation	Interacts with SKIP to form nuclear condensates; modulates AS and cold-responsive gene expression	[13]
bHLH59	*Oryza sativa*	Drought stress	Truncated transcript	OsbHLH59.1/OsbHLH59.2	OsbHLH59.1 promotes growth; OsbHLH59.2 induced by ABA suppresses growth and activates drought response	[12]
CCA1	*Zea mays*	Drought stress	Alternative splicing site	ZmCCA1.1/ZmCCA1.2/ZmCCA1.3	Splice variants responsive to drought and photoperiod; ZmCCA1.1 enhances drought tolerance	[53]
DREB3	*Triticum aestivum*	Drought stress	Alternative 3′/5′ splicing	TaDREB3-I/TaDREB3-II/TaDREB3-III	Only isoform I significantly improves tolerance to drought, salt, and heat in transgenic Arabidopsis	[54]
SRAS1	*Arabidopsis thaliana*	Salt stress	Intron retention	SRAS1.1/SRAS1.2	Salt-induced AS shifts from non-functional SRAS1.2 to SRAS1.1, enhancing CSN5A degradation and salt tolerance	[55]
DHN4	*Cynodon* spp.	Cold/drought/salt stress	Exon skipping	CdDHN4-L/CdDHN4-S	CdDHN4-S enhances ROS scavenging and osmotic response; CdDHN4-L is more effective under cold and drought stress	[56]
HSP70	*Populus trichocarpa*	Heavy metal (Pb) stress	Alternative splicing site	PtHSP70-AS1/PtHSP70-AS2	PtHSP70-AS2 binds Pb(II) 10× more effectively, promotes extrusion and enhances lead tolerance	[7]
EIP1	*Zea mays*	Biotic (fungus) stress	Alternative splicing site	Multiple AS events	Interacts with pathogen effector EtEC81, reprograms 119 AS events to activate immune response	[11]
SR34a	*Solanum lycopersicum*	Biotic (bacteria) stress	Intron retention	Intron retention in defense genes	Effector RipP2 acetylates SR34a, promotes IR in defense genes, suppressing immunity	[57]
MYB6-like	*Malus domestica*	Biotic (fungus) stress	Intron retention	MdMYB6-like-α/MdMYB6-like-β/MdMYB6-like-γ	AS isoform MdMYB6-like-β induced by *Alternaria alternata*, enhances lignin synthesis and disease resistance	[58]

## Data Availability

Not applicable.

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
