# Peer review of "Alternative Splicing of Functional Genes in Plant Growth, Development, and Stress Responses"

_ijms, 2025, doi:10.3390/ijms26125864_

Round 1

Reviewer 1 Report

Comments and Suggestions for Authors

Comment

The review paper presents Alternative Splicing of Functional Genes in Plant Growth, De velopment, and Stress Responses. It discusses alternative splicing (AS), as a crucial post-transcriptional regulatory mechanism  that generates diverse mature transcripts from precursor mRNA, with the resulting functional proteins regulating a wide range of plant life activities. I believe the authors addressed an important topic but it still needs further details and comprehension, which can be provided with incorporating major revisions. 

  1. I suggest to change the keywords into alphabetically orders
  2. I appreciate the figures. But it is better to improve the dpi, quality and the brightness of the figures
  3. In the abbreviation sections, authors only write several abreviations. Please add more as I can see in the whole manuscript, there are so many abbreviations.
  4. I suggest to add a table which listing several genes in this topic. For example Key genes undergoing alternative splicing during plant growth and development. Show the readers the function and any important things. It will help readers because table is easier to be read
  5. How did the author search for literature for this review paper? Authors need to mention it
  6. Line 270, please add” stresses” after temperature
  7. Why authors only elaborate about temperature, drough, and salt stress?why not elaborate another abiotic stresses?
  8. Figure 3, as authors use biorenders to make the figures, please add biorender citation. Also, because authors used many kind of plants as elaboration, it is better to remove “Arbidopsis” words or change it into “ Plant” word.
  9. It is better to add regulatory mechanisms of alternative splicing figures. For example in flowering, response to abiotic stress, and so on,
  10. Line 75, authors said Numerous 73 studies have demonstrated that alternative splicing is widely involved in the transcriptional regulation of higher plants in response to both biotic and abiotic stresses. Why authors did not mention biotic stress in line 269-344?

Author Response

#Reviewer 1:

The review paper presents Alternative Splicing of Functional Genes in Plant Growth, De velopment, and Stress Responses. It discusses alternative splicing (AS), as a crucial post-transcriptional regulatory mechanism  that generates diverse mature transcripts from precursor mRNA, with the resulting functional proteins regulating a wide range of plant life activities. I believe the authors addressed an important topic but it still needs further details and comprehension, which can be provided with incorporating major revisions. 

Comment 1. I suggest to change the keywords into alphabetically orders

Response 1: Thank you for the suggestion. The keywords have been rearranged in alphabetical order in the revised manuscript at line 30.

Comment 2. I appreciate the figures. But it is better to improve the dpi, quality and the brightness of the figures

Response 2: Thank you for your suggestion. We have improved the DPI, brightness, and overall quality of all figures, and the updated figures have been uploaded in the revised manuscript.

Comment 3. In the abbreviation sections, authors only write several abreviations. Please add more as I can see in the whole manuscript, there are so many abbreviations.

Response 3: Thank you for pointing this out. We have carefully reviewed the manuscript and added all relevant abbreviations to the Abbreviation section.

Comment 4. I suggest to add a table which listing several genes in this topic. For example Key genes undergoing alternative splicing during plant growth and development. Show the readers the function and any important things. It will help readers because table is easier to be read

Response 4: Thank you for the helpful suggestion. We have added a new table (Table 1) listing key genes undergoing alternative splicing during plant growth and development, along with their functions and relevant information in line 274.

Comment 5. How did the author search for literature for this review paper? Authors need to mention it

Response 5: Thank you for the comment. We conducted a comprehensive literature search using databases such as PubMed, Web of Science, and Google Scholar. We limited the search to articles published in English, focusing mainly on research from the last 5 years, with some important older references included for background. Articles were screened by titles and abstracts first, then full texts were reviewed to select relevant studies. The search strategy and criteria have now been clearly described in the revised manuscript in line 67-73.

Comment 6. Line 270, please add” stresses” after temperature

Response 6: Thank you for the suggestion. We have added the word “stresses” at line 278 as recommended.

Comment 7. Why authors only elaborate about temperature, drough, and salt stress?why not elaborate another abiotic stresses?

Response 7: Thank you for this insightful comment. We focused on temperature, drought, and salt stresses as they are the most extensively studied and impactful abiotic stresses in current research on alternative splicing in plants. However, we agree that other abiotic stresses are also important, so we have added a brief discussion about other abiotic stresses such as heavy metal stress and other biotic stresses such as pathogen stress or insect pest stress in the revised manuscript in line 369-409.

Comment 8. Figure 3, as authors use biorenders to make the figures, please add biorender citation. Also, because authors used many kind of plants as elaboration, it is better to remove “Arbidopsis” words or change it into “ Plant” word.

Response 8: Thank you for the valuable suggestion. We have added the appropriate citation for BioRender in the figure legend of Figure 3. Additionally, we have replaced the word “Arabidopsis” with “Plant” as recommended in Figure 3.

Comment 9. It is better to add regulatory mechanisms of alternative splicing figures. For example in flowering, response to abiotic stress, and so on,

Response 9: Thank you for the helpful suggestion. We have added several examples of gene regulatory mechanisms related to alternative splicing, including those involved in flowering and abiotic/biotic stress responses in Figure 3 and 4.

Comment 10. Line 75, authors said Numerous 73 studies have demonstrated that alternative splicing is widely involved in the transcriptional regulation of higher plants in response to both biotic and abiotic stresses. Why authors did not mention biotic stress in line 269-344?

Response 10: Thank you for pointing this out. We have added descriptions and discussions on biotic stresses and their involvement in alternative splicing to the revised manuscript in line 382-409.

Reviewer 2 Report

Comments and Suggestions for Authors

This research is interesting and the methodology is well matched with the objectives (the objectives should be better explained in the context of the state of the art). The discussions should be extended in order to add explanations regarding the contributions. Future work should be better related to the limits of this research. 

Author Response

Comment 1. This research is interesting and the methodology is well matched with the objectives (the objectives should be better explained in the context of the state of the art). The discussions should be extended in order to add explanations regarding the contributions. Future work should be better related to the limits of this research. 

Response 1: Thank you for your positive comments and constructive suggestions. We have revised the manuscript, added more detailed descriptions about alternative splicing events in different plant species to better highlight our research objectives. We have also included a discussion of the limitations of our study and provided an outlook on future research directions in the revised manuscript in line 440-469.

Reviewer 3 Report

Comments and Suggestions for Authors

This is a review of a review manuscript, "Alternative Splicing of Functional Genes in Plant Growth, Development, and Stress Responses".

The manuscript is properly written and organized to provide information about alternative splicing genes associated with most plants' physiological traits and stress responses.

The manuscript is good for publication for an audience working in this field of biology. In my opinion, the main idea of this manuscript seems to be introducing genes on this topic, but it might be better to introduce more details about the mechanism by which such splicing variants are applicable to physiological consequences. 

Also, I think it might be great if the authors could include more details on future directions with more specific schemes of possible further studies. 

Author Response

Comment 1. This is a review of a review manuscript, "Alternative Splicing of Functional Genes in Plant Growth, Development, and Stress Responses".

The manuscript is properly written and organized to provide information about alternative splicing genes associated with most plants' physiological traits and stress responses.

The manuscript is good for publication for an audience working in this field of biology. In my opinion, the main idea of this manuscript seems to be introducing genes on this topic, but it might be better to introduce more details about the mechanism by which such splicing variants are applicable to physiological consequences. 

Also, I think it might be great if the authors could include more details on future directions with more specific schemes of possible further studies. 

Response 1: Thank you for your positive evaluation and valuable suggestions. We have expanded the manuscript and added case studies linking specific alternative splicing events to phenotypic traits such as flowering time and stress tolerance in line 145-173,181-192, 200-206, 209-216,225-232, 300-317, 321-328, 345-409. Additionally, new Figure 3, Figure 4 and Table1 have been included to summarize key regulatory pathways mediated by alternative splicing and their physiological effects on plants.

We have also added a dedicated section about future perspectives proposing specific research plans, including developing high-precision splicing regulation tools, multi-omics approaches combining epigenetics and transcriptomics, and integrating computational modeling with biology approaches related to traits such as drought resistance for crop improvement in line 440-447,452-469.

Reviewer 4 Report

Comments and Suggestions for Authors

I wish to thank the authors for this interesting manuscript. However, I have some comments, please, find them below:

1. Sections 1 and 2 are very heavy scientifically, but provide no new information. Specifically, section 2 refers to mostly old papers.

Section 2.2.

2. This manuscript is about plants, and description of yeast and human systems (especially brief and without context and proper comparison to plants) will not fit here. There are some other published papers (some of them cited) critically comparing AS systems from different species, so no need to repeat it here; it'll only add more complexity but not clarification

Sections 3 and 4

3. A lot of old papers. No actual underlying molecular mechanisms were analysed. Just a list of genes that were shown to undergo AS and briefly describe what these genes are going.

4. As a way to improve, I'd suggest removing all old and unrelated papers and focusing on recent (3-5 years old) with the major focus on the new data on the molecular mechanisms of AS itself and the underlying regulatory network. Sections 1 and 2 are extremely complex and hard to follow, while sections 3 and 4 are very brief and simplified. Accordingly, AS molecular mechanisms and associated regulatory pathways should be critically analysed in the discussion.

Author Response

I wish to thank the authors for this interesting manuscript. However, I have some comments, please, find them below:

Comment 1. Sections 1 and 2 are very heavy scientifically, but provide no new information. Specifically, section 2 refers to mostly old papers.

Response 1: Thank you for your valuable feedback. We have revised Sections 1 and 2 to improve clarity and focus, streamlined the content and updated the references by including more recent studies in line 88-90, 107-128.

Section 2.2.

Comment 2. This manuscript is about plants, and description of yeast and human systems (especially brief and without context and proper comparison to plants) will not fit here. There are some other published papers (some of them cited) critically comparing AS systems from different species, so no need to repeat it here; it'll only add more complexity but not clarification

Response 2: Thank you for pointing this out. We have removed the descriptions of yeast and human alternative splicing systems and revised the section. The related descriptions have also been added in line 145-173, 181-192, 200-216,225-232, 285-290, 300-317, and 346-409.

Sections 3 and 4

Comment 3. A lot of old papers. No actual underlying molecular mechanisms were analysed. Just a list of genes thatwere shown to undergo AS and briefly describe what these genes are going.

Response 3: Thank you for pointing out the issue. We have updated the references by removing some irrelevant or outdated ones and have focused on incorporating recent studies from the past five years.

Comment 4. As a way to improve, I'd suggest removingall old and unrelated papers and focusing on recent (3-5 years old) with the major focus on the new data on the molecular mechanisms of AS itself and the underlying regulatory network. Sections 1 and 2 are extremely complex and hard to follow, while sections 3 and 4 are very brief and simplified. Accordingly, AS molecular mechanisms and associated regulatory pathways should be critically analysed in the discussion.

Response 4: Thank you for your systematic suggestions. We have adjusted the length and depth of each section by streamlining Sections 1 and 2, and enriching Sections 3 and 4 with the latest research and mechanistic insights in line 39-60, and 107-128,. We have added the new Figure 3, Figure 4 and Table1 to summarize key regulatory pathways mediated by alternative splicing and their physiological effects on plants in line 271, 276, and 410. We have also expanded the discussion section in line 440-447,452-469.

Round 2

Reviewer 1 Report

Comments and Suggestions for Authors

Thank you for your revision. I really appreciate it.

Author Response

Comments: Thank you for your revision. I really appreciate it.

Response: Thank you very much for your valuable and constructive comments. We greatly appreciate the time and effort you have invested in reviewing our manuscript and providing thoughtful suggestions.

Reviewer 4 Report

Comments and Suggestions for Authors

I wish to thank the authors for implementing changes. However, my comments were mostly left unanswered:

1. Still, I see no new information. Papers cited in sections 1 and 2 are reviews (but now they are recent). However, why would one wish to read this manuscript (a review of other reviews) if cited papers provide more value?

      Comment 1. Sections 1 and 2 are very heavy scientifically, but provide no new information.              Specifically, section 2 refers to mostly old papers.

     Response 1: Thank you for your valuable feedback. We have revised Sections 1 and 2 to                   improve clarity and focus, streamlined the content and updated the references by including             more recent studies in line 88-90, 107-128.

2. Again. This manuscript is about plants. Some cited papers (1 for example is heavily cited, and many others) have NOTHING to do with plants.

       Section 2.2.

      Comment 2. This manuscript is about plants, and description of yeast and human systems            (especially brief and without context and proper comparison to plants) will not fit here. There          are some other published papers (some of them cited) critically comparing AS systems from      different species, so no need to repeat it here; it'll only add more complexity but not clarification

       Response 2: Thank you for pointing this out. We have removed the descriptions of yeast and        human alternative splicing systems and revised the section. The related descriptions have               also been added in line 145-173, 181-192, 200-216,225-232, 285-290, 300-317, and 346- 409.

3. Again. Please, check the literature and remove all papers not related to plants (old and recent).

       Comment 4. As a way to improve, I'd suggest removingall old and unrelated papers and                   focusing on recent (3-5 years old) with the major focus on the new data on the molecular              mechanisms of AS itself and the underlying regulatory network. Sections 1 and 2 are                        extremely complex and hard to follow, while sections 3 and 4 are very brief and simplified.                Accordingly, AS molecular mechanisms and associated regulatory pathways should be                     critically analysed in the discussion.

         Response 4: Thank you for your systematic suggestions. We have adjusted the length and depth of           each section by streamlining Sections 1 and 2, and enriching Sections 3 and 4 with the latest                       research and mechanistic insights in line 39-60, and 107-128,. We have added the new Figure 3,                 Figure 4 and Table1 to summarize key regulatory pathways mediated by alternative splicing and             their physiological effects on plants in line 271, 276, and 410. We have also expanded the discussion             section in line 440-447,452-469.

4. The Table for some reason, was not modified. The Table should provide information about the functional role of AS versions of genes, not a simplified explanation of the gene’s function.

5. Figures were also not modified. While they represent AS versions better, and in some cases depict different functions of AS versions, I do not see it for every presented gene.

Author Response

  1. Still, I see no new information. Papers cited in sections 1 and 2 are reviews (but now they are recent). However, why would one wish to read this manuscript (a review of other reviews) if cited papers provide more value?

 Response 1: Thank you for your comment. We acknowledge that Sections 1 and 2 mainly provide a general overview of the characteristics and basic principles of alternative splicing (AS). These sections serve as a necessary foundation to introduce key concepts, terminologies, and the biological relevance of AS, which are essential for readers. These sections are not meant to present novel information, but rather to set the stage for the main focus of the manuscript in the latter sections. We understand that you do not agree with some of our citation choices, we have revised the manuscript and removed the references. In the revised version of the manuscript, we have merged Sections 1 and 2 into a single introductory section that provides a concise overview of AS in line 29-73.

  1. Again. This manuscript is about plants. Some cited papers (1 for example is heavily cited, and many others) have NOTHING to do with plants.

Response 2: Thank you for pointing this out. We fully acknowledge that the focus of this manuscript is on plant systems. In response to your comment, we have carefully reviewed all citations and removed those that are not directly relevant to plant research.

  1. Again. Please, check the literature and remove all papers not related to plants (old and recent).

Response 3:

Thank you for your suggestion. We agree that the primary focus of this review should remain on plant systems. Accordingly, we have re-evaluated all cited references and removed those unrelated to plants where possible. In a few specific cases, we hope to retain non-plant studies that provide critical mechanistic insights into alternative splicing regulation—particularly where such mechanisms are conserved and help inform understanding of similar processes in plants. While in response to your suggestion, we have removed all papers not related to plants.

  1. The Table for some reason, wasnot modified. The Table should provide information about the functional role of AS versions of genes, not a simplified explanation of the gene’s function.

Response 4: Thank you for your valuable suggestion. Table1 was newly added in response to Reviewer 1's suggestion, as the original version of the manuscript did not contain a table. According to Reviewer 1, we intended to offer a concise summary of key genes and their alternative splicing-related functions. While in response to your suggestion,

we have added more detailed functional annotations of specific AS isoforms and revised the table in line 226.

  1. Figures were also not modified. While they represent AS versions better, and in some cases depict different functions of AS versions, I do not see it for every presented gene.

Response 5: Thank you for your thoughtful comment. Figure 3 and Figure 4 aim to illustrate representative examples of AS contributing to diverse physiological responses through extensive downstream pathways. One gene was randomly selected for each process as an illustrative example to visualize the functional relevance of AS, while Table 1 provides a more comprehensive summary of genes and AS events discussed in this study.

We agree that not all the described genes in this manuscript are presented in the figure, and the primary purpose of these figures is to provide illustrative examples of how AS can generate transcript and protein diversity in plants, with different AS isoforms often performing distinct functions. And the examples included are representative of the regulatory potential of AS in plants. We hope these example-based illustrations serve as a visual aid to highlight the complexity and potential functional outcomes of AS, rather than an exhaustive catalog of all known AS-regulated genes and isoforms, which have been comprehensively covered in Table 1.

Since AS regulation differs greatly among genes and conditions, our intention was to provide diverse, illustrative cases rather than an exhaustive depiction in figures. For instance, some genes included in Figure 4, such as CBF transcription factors involved in cold stress responses, and ZmEIP1 and RipP2 related to biotic stress, are known to broadly influence downstream genes through regulation of the spliceosome. Other examples highlight how sequence variations within the genes themselves lead to AS changes, thereby affecting transcript diversity and function in a more gene-specific manner. These and additional representative genes are described in both the main text and Table 1; the figures present a subset of all genes described in this study for visual emphasis. According to your opinion, we have added clarifying notes to the figure legends and main text in line 216-225, and 360-367.

Round 3

Reviewer 4 Report

Comments and Suggestions for Authors

I wish to thank the authors for implementing changes. However, I still have some minor comments:

1. Please, add some improvements to the figures. Figure 3 is informative and detailed. However, the central image “Plant life cycle” is rather useless – it doesn’t correspond to the used figures (fruit ripening, seed dormancy, and so on). In my view, it’d be better to arrange it in a block (A, B, C and so on) without a central cycle image.

2. PIN7 subfigure (Figure 3) is not informative. Please, arrange it in a way reflecting different functions of AS versions.

3. The same stands for figure 4 – the central image (Arabidopsis plant) is not related to the context.

4. The functions of different AS variants are not clear in the subfigure “Heat stress” and “Heavy metal” (Figure 4). Please, introduce some explanation to the figure, or to the legend.

Please, keep in mind that readers should be able to understand figures separately from the main text.

Author Response

Comment 1: Please, add some improvements to the figures. Figure 3 is informative and detailed. However, the central image “Plant life cycle” is rather useless – it doesn’t correspond to the used figures (fruit ripening, seed dormancy, and so on). In my view, it’d be better to arrange it in a block (A, B, C and so on) without a central cycle image.

Response 1: Thank you for your constructive suggestion. Following your recommendation, we have removed the Arabidopsis image, restructured the figure and added more details in the figure legend in line 221-238. We hope this revision satisfactorily addresses your comment.

Comment 2: PIN7 subfigure (Figure 3) is not informative. Please, arrange it in a way reflecting different functions of AS versions.

Response 2: Thank you for your valuable feedback. We have redesigned the PIN7 subfigure and added more details in the figure legend in line 225-228. We hope these revisions address your concerns.

Comment 3: The same stands for figure 4 – the central image (Arabidopsis plant) is not related to the context.

Response 3: Thank you for pointing this out. We have removed the central Arabidopsis plant image in Figure 4, restructured the figure into individual panels, and expanded the figure legends to provide clearer and more detailed explanations for each panel in line 377-397.

Comment 4: The functions of different AS variants are not clear in the subfigure “Heat stress” and “Heavy metal” (Figure 4). Please, introduce some explanation to the figure, or to the legend.

 Response 4: Thank you for your helpful suggestion. We have added schematic indications and brief labels in Figure 4, and expanded the figure legend. We hope these improvements make the figure more  easier to interpret. We sincerely hope that these revisions address your concerns and that the improved figures now meet your expectations.